# Induction of Somatic Embryogenesis in Tamarillo (*Solanum betaceum* Cav.) Involves Increases in the Endogenous Auxin Indole-3-Acetic Acid

**DOI:** 10.3390/plants11101347

**Published:** 2022-05-19

**Authors:** André Caeiro, Sandra Caeiro, Sandra Correia, Jorge Canhoto

**Affiliations:** Center for Functional Ecology, Department of Life Sciences, University of Coimbra, 3000-456 Coimbra, Portugal; andrecaeiro91@gmail.com (A.C.); caeirosandra@hotmail.com (S.C.); sandraimc@ci.uc.pt (S.C.)

**Keywords:** auxins, embryogenic *calli*, gene expression, HPLC, IAA, immunohistochemistry

## Abstract

Somatic embryogenesis (SE) is a complex biological process regulated by several factors, such as the action of plant growth regulators, namely auxins, of which the most physiologically relevant is indole-3-acetic acid (IAA). In tamarillo, an optimized system for induction of SE creates, after an induction process, embryogenic (EC) and non-embryogenic *callus* (NEC). In this work the endogenous levels of auxin along the induction phase and in the *calli* samples were investigated using chemical quantifications by colorimetric reactions and HPLC as well as immunohistochemistry approaches. Differential gene expression *(IAA 11, IAA 14, IAA 17, TIR 1,* and *AFB3*) analysis during the induction phase was also carried out. The results showed that the endogenous IAA content is considerably higher in embryogenic than in non-embryogenic *calli*, with a tendency to increase as the dedifferentiation of the original explant (leaf segments) evolves. Furthermore, the degradation rates of IAA seem to be related to these levels, as non-embryogenic tissue presents a higher degradation rate. The immunohistochemical results support the quantifications made, with higher observable labeling on embryogenic tissue that tends to increase along the induction phase. Differential gene expression also suggests a distinct molecular response between EC and NEC.

## 1. Introduction

Tamarillo, *Solanum betaceum* (Cav.) Sendt. (syn. *Cyphomandra betacea*) is a small (2–4 m high) solanaceous tree indigenous to South America, specifically to the Andean regions of Argentina, Bolivia Chile, Ecuador, and Peru, and currently cultivated around the globe, namely in California and New Zealand [1]. In its natural environment, tamarillo is found between 700 and 2000 m, preferring lower altitudes in colder climates [2]. It is grown for its edible fruits, which can be consumed fresh, incorporated in recipes [3], or used to prepare jams or other types of processed foods or drinks [4]. Moreover, the fruit presents low caloric and high vitamin content, as well as several natural antioxidants with potential therapeutic activities, making it a possible source of interesting secondary metabolites [5,6]. 

Conventional means such as seeds, cuttings, or grafting into wild *Solanum mauritianum* trees can be used for propagation of tamarillo [4]. However, these techniques present problems such as the genetic variability of seeds, the low interspecific hybridization rate in grafting, and phytosanitary problems [7,8]. In this context, biotechnological tools have been an alternative for plant breeding with several in vitro methodologies described for tamarillo cloning such as micropropagation through axillary shoot proliferation [9], organogenesis [10,11], and somatic embryogenesis (reviewed in [12]).

Somatic embryogenesis (SE) can be defined as a process by which a somatic cell or tissue creates a structure that resembles an embryo (somatic embryo) without fecundation [13]. This structure has embryonic characteristics, such as bipolar organization, lack of vascular contact with the parental tissue and, through a series of developmental stages similar to a zygotic embryo, germinates into a plant [14,15]. The first successful attempt of SE in tamarillo was reported in mature zygotic embryos and hypocotyls [16]. In this protocol, embryogenesis was induced using the auxin 1-naphthaleneacetic acid (NAA) with formation of a small *callus* mass followed by differentiation into somatic embryos in a “one-step” system. On the other hand, if the auxin used was either 2,4-Dichlorophenoxyacetic acid (2,4-D) or picloram, the zygotic embryos and young leaf explants produced an embryogenic *callus* that could be successfully maintained by successive subcultures in the same auxin-containing medium [12,17]. Interestingly, on this type of system a second type of *callus* without embryogenic competence (non-embryogenic) is also obtained. Additionally, in both induction systems, the embryogenic yield was greatly increased by the addition of high levels of sucrose (26 mM) to the culture medium [18].

Auxins, particularly 2,4-D, are the main plant growth regulators (PGRs) used in somatic embryogenesis experiments, with most protocols starting with an induction phase in an auxin supplemented medium followed by an embryo maturation phase on auxin-free or auxin-reduced medium [19]. This typical induction scheme, varying the types of synthetic auxins depending on the original explant, has been followed in tamarillo [6,12]. 

The level of endogenous plant growth regulators (PGRs), namely auxins, is considered one of the most important embryogenic controlling factors [20,21]. Auxin gradients are related with the establishment of bilateral symmetry necessary for proper embryo development in both zygotic and somatic embryos [15]. External stimulation by auxin-like molecules is believed to cause an increase in the endogenous levels of this PGR; thus initiating the cell reprograming stages necessary for embryogenesis [13,22]. Specifically, with the increase in endogenous levels of indole-3-acetic acid (IAA), particularly in the early stages of SE, induction has been extensively reported in several model species, such as carrot (*Daucus carota* L.) and *Arabidopsis thaliana* [23]. The auxin mechanism of action has been related with gene expression modulation, and several proteins have been closely linked to this regulation pathway, namely in embryogenesis [20]. The auxin-responsive protein family (*Aux/IAAs*), auxin response factors (ARFs), and the transport inhibitor response 1 protein (*TIR1*) have been identified as the main response factors active during embryogenesis induction [23].

The aim of this work was to analyze the endogenous levels of auxins, namely indole-3-acetic acid (IAA), the main natural auxin, during somatic embryogenesis and in embryogenic and non-embryogenic *callus* of tamarillo, by chemical quantification of this PGR. Furthermore, quantitative studies were applied along the induction protocol to establish the evolution of auxin concentration. In addition, the distribution of IAA in the tissue was investigated through immunofluorescence microscopy. Finally, the expression of the main genes related to auxin response in embryogenesis was investigated along the induction path in order to relate the genetic effects of this regulator with its endogenous concentrations. 

## 2. Results

### 2.1. Somatic Embryogenesis Induction and Callus Proliferation

In order to obtain enough tissue during the induction phase, leaf segments were culture in the presence 20 µM of picloram, presenting and an embryogenic yield of 44%. The IAA levels were investigated on the final stages of induction starting at 8 weeks until 12 weeks (Figure 1A). Additionally, established embryogenic and non-embryogenic *calli* was also used with embryogenic tissue forming a characteristic white compact structure and non-embryogenic forming a mucilaginous, friable *callus*. For the study of embryogenic competence, previously established embryogenic *callus* (EC) and non-embryogenic *callus* (NEC) of both leaf segments (EC1 and NEC1) and zygotic embryos were used (EC2, NEC 2, and EC3) (Figure 1B). 

To confirm the embryogenic competence, embryogenic and non-embryogenic tissue were transferred to development medium, with the embryogenic tissue forming somatic embryos after 4 weeks, whereas the non-embryogenic tissue became necrotic. In this phase, the influence of auxin polar transport was also assayed as described, with the embryogenic *callus* exposed to different concentrations of 2,3,5-Triiodobenzoic acid (TIBA) (Table 1). In the maturation assay, proliferation of proembryogenic mass was observed in all treatments, while concentrations of TIBA higher than 5 µM show a statistic significant decrease in the formation of normal somatic embryos, while the number of abnormally formed embryos shows no specific trend in relation to TIBA concentration. 

### 2.2. IAA Location on the Induction of SE

The later stages of embryogenic induction were taken from the 8th week onward. All the samples presented some autofluorescence observed in the controls without the antibody labeling (Figure 2A); however, it is possible to observe distinct labeled spots on the proembryogenic tissue, increasing in frequency along the induction phase. The embryogenic tissue follows this pattern with diffuse spots with marked presence of IAA. The presence of IAA appears in the cell peripheral zone. With respect to the non-embryogenic tissue, the labeling is less frequent and even absent in some parts of the tissue. 

In terms of quantification (Figure 2B), the integrated density of the samples showed an increasing tendency along the induction phase, although not statistically significant. The embryogenic tissue, however, presents a higher level of labeling in comparison to the non-embryogenic tissue.

IAA quantification by High Performance Liquid Chromatography (HPLC) (Figure 2C) shows tendency of IAA increase in the induction phase starting with a small, statistically insignificant difference between the 8th and 10th weeks of induction (0.004 ± 0.001 and 0.013 ± 0.004 µg IAA/mg.f.w for EC3, respectively), and a marked increase by the 12th week (0.496 ± 0.001 µg IAA/mg.f.w). These values are still comparatively lower to those presented by embryogenic *callus* sub cultivated in the same media. 

### 2.3. IAA Levels Increment in EC 

The previously induced *callus* lines were assayed for IAA by Ehrlich method and HPLC quantification. When assayed by the Ehrlich reaction and statistically analyzed based on type of synthetic auxin present (Figure 3A), the difference between the endogenous auxin levels in NEC 1 and EC 1 is statistically significant, with EC 1 showing the highest values of IAA (10.49 ± 2.51 µg IAA/mg f.w.). In the case of picloram-induced *calli*, the difference between embryogenic (EC2 and EC3) and non-embryogenic *callus* (NEC2) is statistically relevant (*p* < 0.05), with EC2 showing higher IAA values than EC3, although not statistically different at the confidence interval used. Further statistic comparison between embryogenic *calli* (Figure 3B) indicated a significant statistical difference between the embryogenic tissues induced with different synthetic auxins, with 2,4-D apparently leading to a marked increase in the endogenous level of IAA. A similar statistical test on the non-embryogenic tissue (Figure 3C) showed no differences between non-embryogenic tissues, regardless of biological origin. 

IAA degradation analysis (Figure 3D) was carried out in the *calli* samples, through the quantification of the amount of auxin degraded over a known and fixed period of time (60 min). The results are presented in terms of total intracellular protein to have a specific rate of degradation that can be readily compared between samples (Table 2). This quantification shows no clear distinction in the protein content of embryogenic tissue in relation to the auxin used in the induction phase (0.693 ± 0.075 mg/mL for 2,4-D and 0.625 ± 0.071 mg/mL for picloram). Similarly, the non-embryogenic tissue is also not significantly different between 2,4-D and picloram treatments (0.112 ± 0.029 and 0.290 ± 0.067 mg/mL, respectively). However, the comparison of embryogenic and non-embryogenic *callus* shows a significant difference. Interestingly, EC3 shows an intermediary value between embryogenic and non-embryogenic *callus*.

In the case of 2,4-D-induced tissue (Figure 3D), there were significantly higher rates of degradation in the non-embryogenic tissue (NEC 1 = 1.80 × 10^−3^ ± 1.088 × 10^−3^ µg IAA/mg protein.min), a tendency also observed in the case of picloram-induced tissue where the non-embryogenic presented the highest value of IAA degradation (NEC 2 = 3.21 × 10^−3^ ± 3.06 × 10^−4^ µg IAA/mg protein.min). In this case, EC2 and EC3 showed a statistically similar degradation of IAA. 

IAA content in this case was also analyzed by HPLC (Figure 3E). The IAA was identified via UV-visible spectrum and retention time (Figure 4A–C) and quantified taking into account a calibration curve (Figure 4D). In general, the data obtained by this analysis shows the same trends found when the tissue was subjected to IAA quantification by Ehrlich: EC *callus* lines present significantly higher levels of IAA (2.280 ± 0.303 and 0.621 ± 0.175 µg IAA µg IAA/ mg.f.w for EC1 and EC2, respectively) when compared to the respective NEC line. Furthermore, a separate comparison between the embryogenic *calli* (Figure 3F) revealed statistically higher value in 2,4-D-induced *calli* while in non-embryogenic tissues (Figure 3G) the amounts of auxin were not significantly different. Finally, the results obtained for both analytic methods were compared (Figure 3F) showing a good linear fit (R^2^ = 0.9694) and a systematic higher value of concentration given by Ehrlich quantification. 

### 2.4. Auxin-Related Gene Expression Relates with IAA Levels during SE Induction

In terms of gene expression, 3 Aux/IAA genes and 2 auxin intracellular receptors were analyzed. In terms of the first group, there is a general trend of decrease in gene expression along the induction course, with statistically significant differences between the initial explant (leaf segment) and non-embryogenic *calli.* (Figure 5A–C). Furthermore, in *IAA 11* and *IAA 17,* although statistically insignificant, there appears to be a trend in higher expression in EC when compared to NEC. Both auxin response genes assayed, *TIR 1* and *AFB3* (Figure 5D,E), show a higher expression in the initial induction explants (leaf segments) and a statistically insignificant variation along the induction period and between EC and NEC. 

## 3. Discussion

### 3.1. IAA Distribution Is Important for Somatic Embryo Conversion

Indirect in vitro SE induction protocols can be divided in two stages, one in which somatic cells enter in a dedifferentiated cell state and acquire embryogenic potential and another in which these cells evolve into somatic embryos [24]. These two stages are usually applied in vitro by changing the external stimuli, usually stress or PGR [25]. Tamarillo is one of these cases, where the SE process is induced in an auxin-rich medium, and the proembryogenic masses formed during this stage transform into somatic embryos upon transference to an auxin-free medium [12]. 

Early studies have shown, in both zygotic and somatic embryos, that the endogenous auxin content is important to the developmental program of embryos as well as their germination [26,27]. Therefore, the auxin polar transport inhibitor assays made in this work aimed to test whether somatic embryo development of tamarillo was also affected by the mechanism of polar auxin transport, when the proembryogenic masses are transferred to a medium without auxins. In this stage of embryo development, the endogenous auxin is greatly responsible for the organized division and specification of cells, or embryo patterning [28].

Previous studies have shown that TIBA can inhibit somatic embryogenesis even in the presence of strong auxins such as 2,4-D [29]. Several other studies have also demonstrated that TIBA affects the maturation of somatic embryos, particularly in the earlier stages of globular and heart-shaped embryos [26].

The TIBA mechanism of action is based on blocking auxin transport by binding to PIN regulator efflux carriers [30] without directly antagonizing the response cascade triggered by the auxin, with the auxin polar transport being fundamental for the effective response of the tissue. In fact, several studies seem to support the idea that the polarity of cells is achieved by cell-to-cell communication, greatly influenced by auxins [28,31]. 

The results here presented support this hypothesis, that the cell-to-cell communication mediated by auxins is fundamental for the development of somatic embryos, as the high concentrations of TIBA affected the number of somatic embryos formed.

### 3.2. IAA Quantification through SE Induction

Endogenous levels of PGRs are extremely important in the regulation of plant development [32]. Consequently, their quantification has been extensively carried out in several contexts to understand the biochemical and molecular mechanisms underlying different aspects of morphogenesis [33]. In particular, experiments carried out with different species, such as *Coffea canephora* [34] and *Cunninghamia lanceolate* [35], have shown that the levels of IAA or other auxins strongly affect somatic embryo formation and development.

The Ehrlich reagent has been used to measure several indole-containing molecules, from tryptamines to ergoloid compounds [36], and has been optimized for colorimetric quantifications of IAA [37] and, in specific conditions, IAA and indole-3-butyric acid (IBA) [38], also allowing a discontinuous method for determination of IAA degradation as colorimetric reactions can be applied to protein solutions, in specified conditions, to measure auxin degradation and, therefore, indirectly determine the catalytic activity of the enzymes involved in its oxidation, namely IAA oxidase and peroxidases. This type of discontinuous assay to determine the activity of these enzymes has been previously reported [39]. 

Auxins are enzymatically degraded by either oxidation of the side chains by peroxidases or the oxidation of the indole ring by indole-3-acetaldehyde oxidase (EC 1.2.3.7) or IAA oxidase [40]. This degradation is physiologically important because it leads to the permanent inactivation of IAA [41]. 

The results showed the endogenous level of IAA inversely related to the degradation rate, i.e., tissues with lower IAA levels presented the highest levels of IAA degradation (NEC1 and NEC2), while the tissues with high values of endogenous auxin presented the lowest degradation rate. Data have also indicated that the degradation of IAA in the non-embryogenic *callus* is at least partially responsible for the low concentration of endogenous auxin in this tissue, caused by a higher biotransformation rate. In this context, the homeostasis of auxin in embryogenic and non-embryogenic *calli* of tamarillo seems to be related to the degradation pathway of the complex auxin metabolism. This type of attenuation of the auxin signaling system has been described in *A. thaliana*, where the oxidation of auxin by enzymatic systems was unable to generate the activation of certain auxin-responsive genes [42].

The quantifications made by HPLC differed, in absolute values, from those made by the Ehrlich reaction. However, the overall relationship between the two types of *callus* was similar, with the embryogenic *callus* displaying a higher endogenous IAA concentration than the non-embryogenic one. 

Overall, the endogenous IAA level was dependent on the induction phase and the embryogenic competence of the tissue. These differences have been observed in other SE plant models such as carrot (*Daucus carola* L.) [43] and alfalfa *(Medicago sativa*) [44], systems where 2,4-D is also the auxin used to trigger somatic embryogenesis. 2,4-D appears to be necessary not only for initiation of the somatic embryogenesis process but for the maintenance of embryogenic competence in vitro. The mechanism of action is not completely understood, but clearly involves the expression modulation of several genes related to the metabolism of auxins, mainly IAA biosynthesis, degradation, and transport [24]. The role of 2,4-D as an auxin (directly or indirectly) is still in dispute, with some authors hypothesizing that the dedifferentiation process that the cells experience is a response to the stress caused by the herbicide action of this compound [45].

After the recognition of the differences between embryogenic and non-embryogenic *calli*, the auxin kinetics was investigated along with the induction phase of SE in leaf segments. These have shown a greater embryogenic yield than zygotic embryos, the other explant commonly used on induction protocols [2]. Another important factor to consider is that the leaves were gathered from in vitro cloned plants sharing the same genotype, and because of this the genetic variability factors were less determinant, although genetic variations in plants regenerated from embryogenic embryos have been reported [18]. However, by the 12th week the IAA concentration was still about five times lower than that of the embryogenic *calli*. These results are in accordance with other studies of auxin variation during the SE process, namely those of Yang and co-workers [24] who found, in the somatic embryogenesis process of cotton, a profile of endogenous IAA concentration that decreased in the first stages of cell dedifferentiation and increased in the end of the induction phase, with the final embryogenic tissue presenting values 11-fold higher than the initial explants. In the present work, the endogenous level of IAA in the early cell dedifferentiation periods was not possible to determine; however, the results indicated that the induction phase is characterized by a lower level of IAA, while the maintenance of embryogenic competence is largely dependent on high concentrations of IAA. In fact, tissues with minor embryogenic competence (such as the *callus* tissue EC 3) presented a significantly lower concentration of IAA.

Several studies have shown that the endogenous auxins are responsible for the activation of a complex pathway, which leads to the activation of several genes, either related with the metabolism of auxin or metabolic pathways, such as basic metabolic pathways, and the biosynthesis of secondary metabolites [46]. Additionally, Yang and co-workers (2012) [24] also found distinct transcription profiles in embryogenic and non-embryogenic tissues, related with the auxin signal pathway. Comparative proteomic studies in tamarillo have also shown differently expressed proteins in embryogenic and non-embryogenic *callus*, despite the type of auxin used for induction [47].

Given the results presented here, and the analogy to other species, it can be assumed that the endogenous auxin level is influencing the proteome of the different cell lines.

### 3.3. Immunolocalization of IAA during SE Induction

The distribution of auxin in the tissue is an important factor for somatic embryogenesis with several signaling events related with this hormone linked to early somatic embryogenesis [48]. Furthermore, the induction process for tamarillo SE applied in this work gives rise to both embryogenic and non-embryogenic tissue, a result that could be, at least partially, related to the distribution of IAA in the proembryogenic tissue along the induction phase. Our results point to an increase in this level along the induction phase consistent with the chemical quantifications, and a distribution seems to be ubiquitous in the proembryogenic masses and the embryogenic tissue. The only significant distinction can be seen when the embryogenic and non-embryogenic tissue is compared. Similar differences have been reported in other woody plants [49]. Overall, these results seem to indicate that distribution of IAA in the tissues is directly related to the embryogenic competence, with a similar result described for other model species such as *Arabidopsis thaliana* [50]. 

### 3.4. Auxin-Related Gene Differential Expression Is Influenced by IAA Endogenous Levels

The differential genetic expression triggered by auxin or auxin-like stimulation is considered central to almost all plant development processes, namely SE [23]. In this work, as the endogenous IAA levels were shown to be different along the induction process and between EC and NEC lines, gene expression assays were carried out to elucidate some of the molecular aspects of cell response to the embryogenic trigger. Several similar studies have been carried out and show that certain genes are over- or under-expressed in SE protocols and can be related to embryogenic competence [23,51]. The molecular mechanism of auxin regulation mediated by these genes is partially known: in the absence of auxin, the *Aux/IAA* gene family interacts with Auxin Responsive Factor (*ARF*), inhibiting its activity and decreasing auxin response, whereas in auxin presence they are targeted for ubiquitin-mediated degradation [52,53]. These genes, namely *IAA 17*, have been shown to be over-expressed in initial SE stages of *A. thaliana* [53]. In this work the *Aux/IAA* genes (*IAA 11* and *IAA 17*) were found over-expressed in several stages of SE when compared to non-embryogenic *calli*. As these genes are repressors of auxin-induced gene expression [54], it is possible that their higher relative expression is linked with a proper cell response to the highly auxin enriched culture medium resulting in embryogenic competence [55]. Furthermore, the low expression values of NEC *calli* might directly explain the low embryogenic competence that this tissue presents. Previous studies have hinted to this type of response by studying the genetic expression profile of highly responsive cultivars of *Gossypium hirsutum* in comparison to recalcitrant ones [56]. However, these expression levels can also be related to the low concentration and high degradation rates of IAA that have been found in NEC *calli*. In fact, the presence of synthetic auxins has increased the endogenous concentration of IAA in EC *calli*. Therefore, it is possible that a distinct molecular mechanism is responsible for the low concentration of auxins in NEC and this factor is influencing the low expression of *Aux/IAA*. This hypothesis should be further investigated in future works. 

Lastly, the ubiquitin ligase complex responsible for degradation of *Aux/IAA* contains the *TIR1*- and *AFB*-encoded genes and these proteins are considered auxin receptors [57]. These genes were found under-expressed in all tissues when compared to the initial explant used in the induction protocol (leaf segments). Again, this fact might hint to a “low auxin sensitive” environment that is effectively suppressed in EC by a higher level of endogenous IAA that does not exist in NEC; therefore critically influencing the embryogenic competence of both *calli*. This hypothesis, along with the precise molecular mechanism of auxin increase in EC, should be further investigated. 

## 4. Materials and Methods

### 4.1. Somatic Embryogenesis Induction from Leaf Segments of In Vitro Propagated Shoots 

Tamarillo plants (red variety) were used for SE induction, were micropropagated from previously established shoot cultures from in vitro germinated seeds in MS [58] propagation medium supplemented with 8.6 mM sucrose, 0.88 µM of 6-benzylaminopurine (BAP), and 6 g/L of agar (Sigma-Aldrich, St. Louis, MI, USA) and pH was adjusted to 5.6–5.8 before autoclaving. The plants were segmented (1–1.5 cm) and subcultured monthly in the same medium and kept in a growth chamber at 25 °C, in a 16 h photoperiod, at 25–35 μmol m^−2^ s^−1^ (white cool fluorescent lamps). The apical leaves from the clones (2–4 for each plantlet) were aseptically removed, after one month in propagation medium, and used for SE inductions as previously described [12,17]. Briefly, the leaves were segmented (area of approximately 0.25 cm^2^), randomly punctured on the abaxial side and placed on test tubes (15 cm × 2.2 cm) containing approximately 12.5 mL of MS medium supplemented with 26 mM sucrose and a synthetic auxin, 20 µM of picloram. The pH was adjusted to 5.6–5.8 before autoclaving and 2.5 g/L of phytagel^TM^ (Sigma-Aldrich, St. Louis, MI, USA) was added as the gelling agent. All the culture media used were autoclaved at 121 °C for 20 min.

### 4.2. Embryogenic and Non-Embryogenic Calli Subcultures and Maintenance

The induction of proembryogenic *calli* as previously described and its subculture were carried out in the same culture medium in dark conditions in a growth chamber at a temperature of 24 ± 1 °C for 12 weeks. During the later stages of the induction stage (8–12 weeks), samples of dedifferentiating explants from leaf segments were periodically (every 2 weeks) removed from the induction medium and frozen in liquid nitrogen for further analysis. Additionally, for genetic analysis, samples of early-stage dedifferentiation (2 weeks) were also removed from the culture medium and immediately frozen. Unless otherwise stated, all samples were made in triplicate. 

At the end of the induction period (12 weeks), masses of EC were transferred to Petri dishes (90 mm in diameter and 15.9 mm in height) containing 30 mL of a hormone-free embryo development MS medium, supplemented with 11.6 mM sucrose and 2.5 g/L of phytagel^TM^ (Sigma), to evaluate embryogenic competence by the development of somatic embryos. Based on their embryogenic ability two lines of EC were selected: EC2 (high embryogenic ability) and EC3 (low embryogenic ability). Furthermore, to study the influence of auxin gradients in somatic embryo conversion, TIBA was added to the embryo development medium in a concentration between 0.5 and 10 µM. The TIBA solution was sterilized by filtration with a 0.2 µm filter and added to the medium at a temperature of about 60 °C to avoid thermal degradation. An initial mass of about 200 mg of *callus* tissue was used and after 4 weeks of growth in dark conditions at a temperature of 24 ± 1 °C the final mass was registered and number of somatic embryos (morphologically normal and abnormal) counted. The results are presented as a percentage of mass increment ((final mass − initial mass)/initial mass × 100) and number of somatic embryos per gram of tissue. The conversion assays were carried out in quadruplicate.

Additionally, to access the endogenous auxin levels of embryogenic and non-embryogenic masses from other tissues (particularly in terms of the synthetic auxin used), previously induced EC and NEC *calli* form zygotic embryos were used. These were maintained in test tubes (15 × 2.2 cm^2^) containing approximately 12.5 mL of MS medium supplemented with MS medium with 26 mM sucrose and 9 µM of 2,4-D with and 2.5 g/L of phytagel^TM^ (Sigma). These were termed EC 1 and NEC 1, respectively. 

### 4.3. Quantification of IAA 

#### 4.3.1. Ehrlich Reaction

In a first approach, the IAA content in the established *calli* lines (from zygotic embryos, EC 1 and NEC 1, and leaf explants, EC 2, EC 3, and NEC 3) was assayed using the colorimetric method described by Anthony and Street [37]. This methodology was applied to broadly evaluate the IAA endogenous concentration in the proembryogenic masses and evaluate the IAA degradation rates of these *calli* to access if the embryogenic competence and culture medium influenced the endogenous levels of auxins in the same patterns. Ehrlich reagent was prepared by dissolving 2 g of *p*–dimethylaminobenzaldehyde (PDAB, Sigma-Aldrich, St. Louis, MI, USA) in 100 mL HCl 2.5 M. The plant material, on average 600 mg of fresh mass, was ground in a sterilized mortar with K-phosphate buffer 0.01 M (pH 6.0) (1 mL/500 mg.f.w) and centrifuged (4800 g; 20 min). After centrifugation, the supernatant was used for the quantification. The reaction was initiated with successive addition of 2 mL of TCA (100%) (Sigma) and Ehrlich reagent to 1 mL of sample. After an incubation period of 30 min, the absorbance at 530 was measured against a blank solution of K-phosphate in a Jenway 7305 spectrometer. A calibration curve was prepared using buffered solutions of IAA with concentrations between 2 and 50 µg/mL. The results are presented as µg of IAA per mg of fresh tissue (µg/mg.f.w).

To measure the degradation of IAA by the tissue, *callus* samples were treated as before and incubated in IAA solution (0.02 mM IAA; 0.02 mM MnCl2, K-phosphate buffer) for 90 min before the Ehrlich reaction was carried out. The results are presented as µg of IAA degraded per mg of protein per minute (µg IAA/mg protein.min). 

The total protein was assayed using Bio-Rad Protein Assay based on Bradford’s reaction (Bradford, 1976) in a 96-well microplate. A calibration curve was constructed using concentrations of BSA between 5 and 40 µg/mL. All measurements were made simultaneously and in triplicate at 595 nm in a SPECTRAmax PLUS 384 spectrophotometer (Molecular Devices, Sunnyvale, CA, USA).

#### 4.3.2. HPLC

The colorimetric quantifications showed differences in both endogenous levels of auxin and degradation rates; however, this quantification method is described as less sensitive than other methods, namely chromatographic methods such as HPLC, and proved less precise in dedifferentiating leaf segments. Therefore, to have a more sensitive approach that also allowed the analysis of time courses, HPLC analysis was employed. The quantification of IAA by HPLC was based on the method described by Kim and co-workers [59] with modifications. Briefly, the plant material (8, 10, and 12 weeks of induction as well as EC1EC2, EC2, NEC 1, and NEC 2) was extracted in 100% methanol (2.5 mL per gram of fresh weight tissue), IPA was added as an internal standard (10 µg/g.f.w.), and the resulting extract was cleared by centrifugation (16,000× *g*, 10 min) at 4 °C. Before the next steps, the polarity of the extract was increased by adding one volume of pure water. The sample was then extracted by two steps of serial partition against 100% ethyl acetate. In the first, the pH of the aqueous phase was adjusted to higher than 9 (1 M KOH) and after separation of phases by centrifugation (16,000× *g*, 10 min), the aqueous phase was transferred to a new tube and the pH was reduced to less than 3 and again partitioned against ethyl acetate. After separation of phases by centrifugation (16,000× *g*, 10 min), the organic phase was collected, completely dried in vacuum, and dissolved in a minimal volume of 100% methanol, micro-filtered, and injected in the HPLC apparatus.

The samples were then analyzed in an HPLC system composed of a Gilson 234 injector, Gilson 305 pumps, and Waters Spherisorb^®^ 5 µm ODS2 (C18) 4.6 × 250 mm column and a Gilson 170 diode array detector (Gilson, Madison, WI, USA). The compounds were resolved with an isocratic elution similar to that used by Nakurte and co-workers [60], consisting of 56% methanol and 44% water and orthophosphoric acid (pH = 2.3) with a flow rate of 1 mL/min. The system used the control and analysis software Gilson Unipoint v 5.11. The detector wavelength was set at 282 nm. Calibration curves of standard concentrations of both IAA and IPA between 0.5 and 25 µg were prepared in triplicate. Additionally, the resolving power of the isocratic elution was tested with mixtures of both the components in different concentrations. 

### 4.4. Immunohistochemistry IAA Localization

The total quantification of IAA during late induction phase was complemented with IAA immunolocalization studies. For the localization of IAA in specific cells/tissues during SE induction, samples were collected from several time-points during induction from leaf explants (8, 10, and 12 weeks) and from embryogenic and non-embryogenic *callus* tissue previously induced (EC2 and NEC2) also from leaf explants and in the presence of picloram. The samples were subjected to a fixation protocol with an overnight fixation step in cold ethanol:acetic acid 3:1 (*v*:*v*) followed by successive incubations in solutions with increasing amounts of sucrose in PBS buffer (0.01 M, pH 7.4): 10% sucrose for 3 h, 15% sucrose for 3 h, and finally 34% sucrose and 0.01% safranin overnight. Before each incubation the samples were vacuum infiltrated for 15 min. After the fixation process, the samples were frozen in optimal temperature cutting compound (OCT, Sakura^®^ Finetek, Torrance, CA, USA), cut in a cryostat microtome into 14 µm thick sections that were mounted into poly-L-Lysine-coated slides. The sections were then digested in a 2% driselase^®^ (Fluka) solution for 30 min at 37 °C and washed with PBS. After the digestion, the samples were blocked with a 10% (*w*:*v*) bovine serum albumin (BSA, Thermofisher, Waltham, MA, USA) in PBS buffered solution for 1 h. The samples were then stained with 0.01% indole-3-acetic acid polyclonal antibody in 0.3% (*w*:*v*) BSA/PBS buffer for 24 h, washed three times with PBS, and labeled with the secondary antibody 0.002% (*v*:*v*) Alexa fluor^®^ 633 goat anti-rabbit (Thermofisher, Waltham, MA, USA). Sections were then mounted in DakoCytomation fluorescent mounting medium (Abcam, Cambridge, UK) and examined under a confocal microscope (Zeiss LSM510 META; Carl Zeiss, Jena, Germany).

### 4.5. Expression of Auxin-Related Genes

Total RNA was extracted from multiple samples during the SE induction and from maintained *calli*, using the kit NucleoSpin^®^ RNA Plant (MACHEREY-NAGEL GmbH & Co. KG, Duren, Germany) following the manufacturer’s instructions. The final concentration of RNA of each sample was measured using a spectrophotometer (NanoDropTM, Thermo Scientific, MA, USA). RNA quality was confirmed with the A260/A280 and A260/A230 ratios and in an agarose gel electrophoresis. cDNA was produced from 1 μg of total RNA from each sample using NZY First-Strand cDNA Synthesis Kit (NZYTech, Lda.—Genes and Enzymes, Lisbon, Portugal) according to the manufacturer’s instructions.

Quantitative PCR gene expression analysis of three genes coding Aux/IAA proteins (*IAA11*, *IAA14*, and *IAA17*) and two auxin receptors (TRANSPORT INHIBITOR RESPONSE 1, *TIR1* and AUXIN SIGNALING F-BOX 3, *AFB3*) was made using NZYSpeedy qPCR Green Master Mix (2x) (NZYTech, Lda.—Genes and Enzymes, Lisbon, Portugal), following the instructions provided with the samples diluted 50 times. Samples with the mix were pooled in a 96-well qPCR plate and measured in C1000 TouchTM Thermal Cycler (Bio-Rad Laboratories, Lda., Amadora, Portugal). For reliable quantitative PCRs, two reference genes were also chosen in order to normalize the data of Ef1α and IRON SUPEROXIDE DISMUTASE, FeSOD [61]. All the primers (Table 3), with the exception of TIR1 gene primers (designed for *Solanum lycopersicum* GQ370812.1), were designed for *Solanum betaceum* transcript sequences obtained from embryogenic cell RNAseq libraries (data not published), using the NCBI primer design tool.

The expression values (Cq) obtained were first normalized using the mean Cq values for the reference genes used. The method used to analyze the qPCR data was the relative quantification method, or 2^−ΔΔCT^ method, where the ΔΔCT value = (CQ Target—CQ Reference) [62].

### 4.6. Statistical and Data Analysis 

The homogeneity of variances was tested with the Brown–Forsythe test (*p* < 0.05). In the case of homogeneity of variances, the data was analyzed with a one-way analysis of variance (ANOVA) and, where necessary, the means were compared by Tukey test (*p* < 0.05). In the case non-homogenous variances, a Kruskal–Wallis one-way analysis of variance was used and the means compared by Dunn’s multiple comparison test (*p* < 0.05). 

To compare outputs between the two IAA quantification methods, a linear interpolation was used with a 95% confidence interval using the program GraphPad^®^ Prism version 6.1 for windows. 

Immunofluorescent results were analyzed using Fiji software [63], taking the control without antibody labeling for threshold determination and computation of the integrated density of fluorescents.

## 5. Conclusions

Somatic embryogenesis is complex biological process mediated and regulated by several molecular mechanisms. One the most important aspects is the level of endogenous auxin present in the explants. The present work aimed to further study the influence of auxins, namely IAA, in the induction of SE and the conversion of the proembryogenic masses into somatic embryos in indirect somatic embryogenesis of tamarillo, a system that has been extensively optimized and studied. As such, several quantification assays were carried out along the induction phase and on both embryogenic and non-embryogenic *calli*, as well as some differential gene expression studies. 

The results in this work show the kinetics of the endogenous IAA content as increasing along the induction phase and a higher concentration of EC over NEC, as well as a decreased IAA degradation rate in EC. Some changes in gene expression level have also been found in some of the main auxin response genes. Furthermore, an assay with TIBA, an auxin polar transport inhibitor, has shown indirectly that IAA is important of the conversion of EC into somatic embryos.

Altogether, the results hint that IAA endogenous concentration is important in induction of proembryogenic masses as it tends to increase along the dedifferentiation process. Additionally, it appears to be relevant for the acquisition and maintenance of embryogenic competence as EC systematically presented a higher concentration of NEC. The genetic expression results presented a general difference between the initial explants of SE and the final proembryogenic masses. Interestingly, the expression values are not quantitatively different between both types of *calli* assayed, a fact that might suggest a different molecular mechanism, either at the biosynthetic or degradation steps in the complex metabolic pathway of auxin homeostasis. Therefore, future studies should aim to further characterize the biosynthetic of IAA in the *calli* as well as a deeper molecular characterization.

## Figures and Tables

**Figure 1 plants-11-01347-f001:**
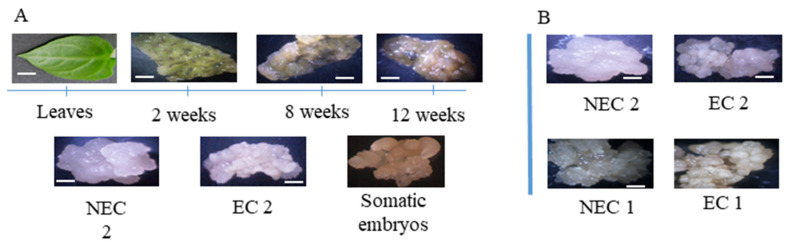
Plant material. (**A**) Time course of leaf explant SE induction; leaves from *in vitro* established plantlets were used. EC 2 and NEC 2 are the embryogenic and non-embryogenic *calli* resulting from the induction process. (**B**) EC 2 and NEC 2 were previously established from zygotic embryos and used in the IAA quantifications. The bars in each figure represent 1 mm with exception of the leaves in which the length is 1 cm.

**Figure 2 plants-11-01347-f002:**
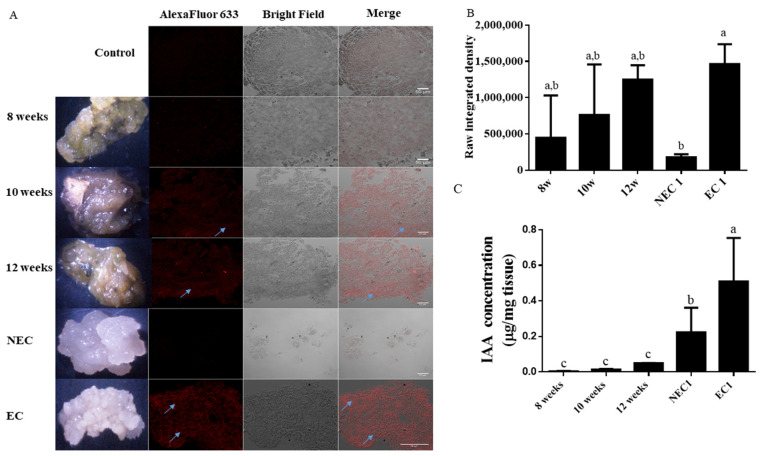
IAA assays in time courses from leaf segment induction. NEC and EC are non-embryogenic and embryogenic *calli*, respectively, similar to EC 1 and NEC 1 represented in Figure 1. (**A**) Immunohistochemistry observations. Fluorescence, transmission, and composed image of the different tissues, unlabeled tissue samples from EC were used as control (C—control; 8 w—8 weeks induction; 10 w—10 weeks induction; 12 w—12 weeks induction; NEC—non-embryogenic *callus*; EC—embryogenic *callus*). (**B**) Raw integrated intensity for each sample. (**C**) IAA quantification by HPLC. Results are presented as mean ± SD. Different letters are significantly different according to Tukey test (*p* < 0.05).

**Figure 3 plants-11-01347-f003:**
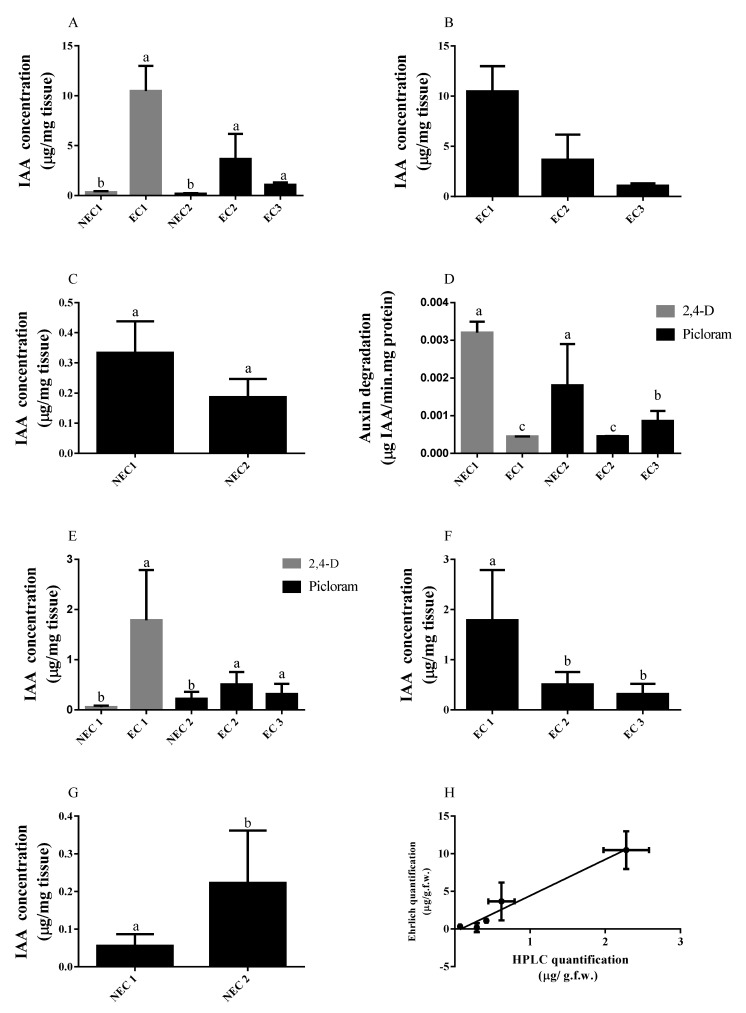
IAA quantification on induced embryogenic (EC) and non-embryogenic (NEC) *calli*, where EC 1 and NEC 1 are induced from zygotic embryos in 2,4-D supplemented medium and NEC 2, EC 2, and EC 3 are from leaf segments in a picloram-supplemented medium. (**A**) IAA quantification by Ehrlich reaction. (**B**) Statistical analysis of IAA in embryogenic *calli* assayed by Ehrlich reaction. (**C**) Statistical analysis of IAA in non-embryogenic *calli* assayed by Ehrlich reaction. (**D**) IAA degradation measured by discontinuous assay. (**E**) IAA quantification by HPLC. (**F**) Statistical analysis of IAA in embryogenic *calli* assayed by HPLC. (**G**) Statistical analysis of IAA in non-embryogenic *calli* assayed by HPLC. (**H**) Comparison of IAA measurement Ehrlich reaction and HPLC comparison between HPLC and Ehrlich reaction. Results of HPLC quantification (x axis) were plotted against the results of Ehrlich quantification (y axis) for each *callus* tissue tested. The linear fit equation is y = 4.818x − 0.3963 (R^2^ = 0.969). Results are presented as mean ± SD. Different letters are significantly different according to Tukey test or by the unpaired *t*-test in the case of non-embryogenic *calli* analysis (*p* < 0.05; *n* = 3).

**Figure 4 plants-11-01347-f004:**
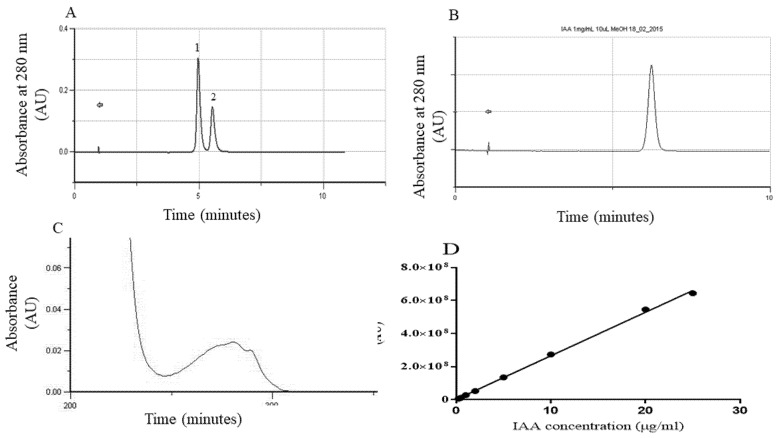
HPLC quantification parameters. (**A**) Retention time of IPA (peak 1, Rt = 5.34 min) and IAA (peak 2, Rt = 6.00 min) in a sample. (**B**) IAA standard (Rt = 5.98 min). (**C**) UV-spectrum of IAA. (**D**) Calibration curve used in the quantification of IAA with the linear equation Area = 2.633 × 10^7^[IAA] + 2.094 × 10^6^ (R^2^ = 0.9984).

**Figure 5 plants-11-01347-f005:**
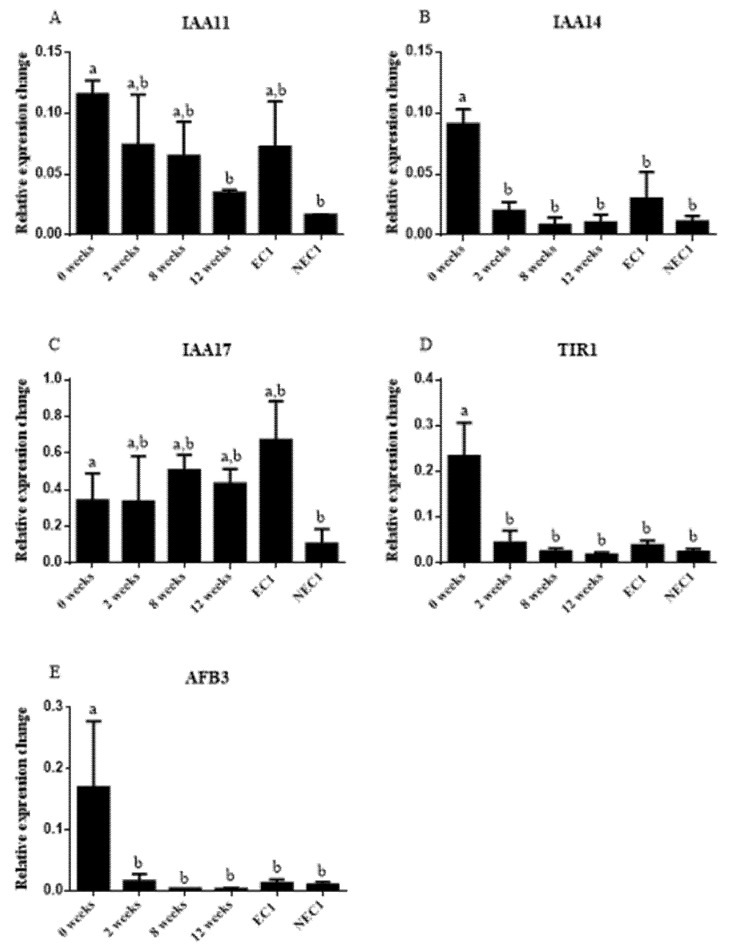
Relative gene expression. Relative expression was calculated in relation to reference genes previously validated for tamarillo. Results are presented as mean ± SD. Different letters are significantly different according to Tukey test (*p* < 0.05).

**Table 1 plants-11-01347-t001:** Results of the auxin polar transport inhibitors assay. Values are presented as mean ± SD (*n* = 4); values in the same column with different letters are statistically different by Tukey test (*p* < 0.05) in the case of the mass increment, while for abnormal embryo formation the letters refer to Dunn’s multiple comparison test.

TIBA (µM)	Mass Increment (g)	Embryo Formation (Number/g of Tissues)	
		Normal	Abnormal
0	1.057 ± 0.140 ^a,b^	22.16 ± 9.58 ^a^	9.08 ± 8.02 ^b^
0.5	1.140 ± 0.201 ^a^	17.42 ± 1.99 ^a^	3.86 ± 1.18 ^b,c^
1	1.132 ± 0.102 ^a^	14.4 ± 3.45 ^a^	1.71 ± 0.20 ^c^
5	1.045 ± 0.217 ^b^	7.77 ± 2.90 ^b^	10.13 ± 2.62 ^a,b^
10	0.695 ± 0.190 ^b^	2.08 ± 1.50 ^b^	1.51 ± 0.744 ^c^

**Table 2 plants-11-01347-t002:** Total protein content in the different *calli*. Results are presented as mean ± SD (*n* = 3). Different letters are significantly different according to Tukey test (*p* < 0.05).

Tissue	Protein Concentration (mg/mL) ± SD
NEC1	0.112 ± 0.029 ^c^
NEC2	0.290 ± 0.067 ^c^
EC1	0.693 ± 0.075 ^a^
EC2	0.625 ± 0.071 ^a^
EC3	0.388 ± 0.290 ^a,b^

**Table 3 plants-11-01347-t003:** Primers used for the gene expression assay.

Gene	Forward Primer	Reverse Primer	Amplicon Length (bp)
*Ef1α*	ACAAGCGTGTCATCGAGAGG	TGTGTCCAGGGGCATCAATC	183
*FeSOD*	TCACCATCGACGTTTGGG AG	GACTGCTTCCCATGACACCA	114
*IAA11*	AGGAAGGGTGCCTAGTTAGC	TGACACCCCTCGAGTAAGGA	631
*IAA14*	AGTTTTCCGACGAAGAGGGT	GTTGGCCACCAGTGAGATCAT	332
*IAA17*	TTGATGAAGAGCTCGGAGGC	CCCCGTGGCCTTATTTACGA	335
*TIR1*	AGATGGCTGTCCAAAGCTCC	GAGCCTTGTCTCCAAACGGA	389
*AFB3*	CTGTACGGAAATGGGGTGCT	GCAGAGTACGGGGAACCAAA	284

## Data Availability

Not applicable.

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
