# Peer review of "Induction of Somatic Embryogenesis in Tamarillo (*Solanum betaceum* Cav.) Involves Increases in the Endogenous Auxin Indole-3-Acetic Acid"

_plants, 2022, doi:10.3390/plants11101347_

Round 1

Reviewer 1 Report

I have read the manuscript „Induction of somatic embryogenesis in tamarillo (Solanum betaceum Cav.) involves localized increments in the endogenous auxin indole-3-acetic acid” prepared by Andre Caeiro et al. for publication in Plants MDPI.

Although generally the experiments were good performed and the results have scientific value and importance, in my opinion the description of the applied methodology, and the way, in which results are presented and interpreted,  need major revision. In the present form the manuscript is unsuitable for publication. Some very important methodological informations are missing, and the interpretation of results seems to be very often incompatibile with the statistical analysis.

I am sending the pdf file with the manuscript, in which all my detailed comments and suggestions are written. The Authors should obligatory follow them to improve the manuscript.

Author Response

Dear Editor,

Firstly, we would like to acknowledge the suggestions, corrections and comments of the reviewers that allowed us to improve the manuscript.

            All the comments have been taken into account and several sections of the manuscript have been rewritten, expanded or new figures have been added. The original manuscript lists the changes that we have made in response to the reviewers and in this letter we respond to the comments and corrections of each reviewer. Furthermore, as a new discussion topic was added, and another expanded, the reference list was also modified.

            In the following text our answers to the different reviewers are in blue.

            We hope that the new manuscript has met the standard for publication in Plants.

            Kind regards,

            Jorge Canhoto (on behalf of all authors)

Reviewer 1

  1. Title

“Localized” has been deleted”

  1. Abstract

The gene abbreviations have been added

  1. Introduction

The introduction has been changed (now in lines 49-50) to clarify the meaning of the sentence.

  1. Results
    1. Figure 1 - Caption has been expanded and identification of EC and NEC calli
    2. Table 1 – Letters referring to statistical analysis have been corrected and the text changed accordingly.
    3. Figure 2 – Caption has been expanded to explain the tissues used in the analysis.
    4. Figure 3 – Statistical analysis previously only mentioned in the text has been added; legend and text changed accordingly.
    5. Table 2 – Letters referring to statistical analysis have been added.
    6. The result presentation for the genetic results has been rewritten.
  2. Discussion
    1. Latin names have been provided.
    2. Italicization has been added where necessary.
    3. Discussion concerning the gene data has been extended.
  3. Methods
    1. In vitro propagation section has been extensively rewritten with more detail as well as the origin of the embryogenic and non embryogenic calli
    2. It is explicitly stated each samples were used for each analysis as well as the replica of each sample.
    3. Culture vessels used are described.
    4. The suppliers of the more unusual chemicals were provided
  4. Conclusions
    1. The conclusion section has been expanded.

Reviewer 2 Report

The authors of the manuscript “Induction of somatic embryogenesis in tamarillo (Solanum betaceum Cav.) involves localized increments in the endogenous auxin indole-3-acetic acid” attempt to detect and quantify the endogenous levels of indole-3-acetic acid in embryogenic and non-embryogenic callus of Solanum betaceum. The results can explain the involvement of endogenous indole-3-acetic acid in somatic embryo formation in tamarillo.

Please address the following comments:

L47: (globular, heart-shaped, torpedo, and cotyledonary). What about monocotyledons?

L52: 2,4-Dichlorophenoxyacetic acid

L70: Please expand the abbreviation at first use. ‘IAA’

L79: Please delete: induction.

L94: callus (non-italic).

L95: Please expand the abbreviations: EC and NEC.

L97: Please enlarge Figure 1 (improve the quality).

L98: somatic embryos? Please replace Figure 1A (Somatic embryos), showing several somatic embryos.

L105: Please expand the abbreviation: TIBA.

L105-109: Please discuss the results (missing in discussion).

L127: Figure 2: Please include different stages of somatic embryogenesis (C – control, 8 w – 8 weeks induction, 10 w – 10 weeks induction, 12 w – 12 weeks induction, NEC – Non- embryogenic callus, EC – Embryogenic callus) for better understanding. Also, improve the quality of the figures.

L132: Please provide the HPLC chromatograms of the IAA standard and samples. ‘IAA quantification by HPLC’

L175: Table 2. Statistical analysis is required.

L199: Italicize the genes: IAA 17 and IAA 14 (throughout the text).

L200: Italicize the genes: TIR 1 and AFB3 (throughout the text).

L214: Coffea canephora (Italic).

L215: somatic embryo instead of somatic embryogenesis.

L238: A. thaliana (Italic).

L297: Arabidopsis thaliana (Italic).

L465: calli (non-italic).

Author Response

Dear Editor,

Firstly, we would like to acknowledge the suggestions, corrections and comments of the reviewers that allowed us to improve the manuscript.

            All the comments have been taken into account and several sections of the manuscript have been rewritten, expanded or new figures have been added. The original manuscript lists the changes that we have made in response to the reviewers and in this letter we respond to the comments and corrections of each reviewer. Furthermore, as a new discussion topic was added, and another expanded, the reference list was also modified.

            In the following text our answers to the different reviewers are in blue.

            We hope that the new manuscript has met the standard for publication in Plants.

            Kind regards,

            Jorge Canhoto (on behalf of all authors)

Reviewer 2

  1. L47: (globular, heart-shaped, torpedo, and cotyledonary). What about monocotyledons?

The text has been changed to make it more general, it refers now that “ the structure (…) through a series of developmental stages similar to a zygotic embryo, germinates into a plant.

  1. L52: 2,4-Dichlorophenoxyacetic acid

Changed accordingly.

  1. L70: Please expand the abbreviation at first use. ‘IAA’

“Indole-3-acetic acid” has been added.

  1. L79: Please delete: induction.

Changed accordingly.

  1. L94: callus (non-italic).

Changed accordingly.

  1. L95: Please expand the abbreviations: EC and NEC.

“Embryogenic Callus and Non-embryogenic Callus” has been added.

  1. L97: Please enlarge Figure 1 (improve the quality).

The Figure has been enlarged.

  1. L98: somatic embryos? Please replace Figure 1A (Somatic embryos), showing several somatic embryos.

The image of the single somatic embryo has been replaced with an image showing several somatic embryos emerging from an embryogenic callus.

  1. L105: Please expand the abbreviation: TIBA.

“2,3,5-triiodobenzoic acid” has been added.

  1. L105-109: Please discuss the results (missing in discussion).

A new discussion topic “3.1. IAA distribution is important in somatic embryo conversion” was added.

  1. L127: Figure 2: Please include different stages of somatic embryogenesis (C – control, 8 w – 8 weeks induction, 10 w – 10 weeks induction, 12 w – 12 weeks induction, NEC – Non- embryogenic callus, EC – Embryogenic callus) for better understanding. Also, improve the quality of the figures.

The figure has been changed.

  1. L132: Please provide the HPLC chromatograms of the IAA standard and samples. ‘IAA quantification by HPLC’

A new figure (figure 4) has been added with the required information. Also, the standard curve and equation used are presented.

  1. L175: Table 2. Statistical analysis is required.

Statistical analysis is now provided, and the caption of the table has been changed accordingly.

  1. L199: Italicize the genes: IAA 17 and IAA 14 (throughout the text).

Changed accordingly.

  1. L200: Italicize the genes: TIR 1 and AFB3 (throughout the text).

Changed accordingly.

  1. L214: Coffea canephora (Italic).

Changed accordingly.

  1. L215: somatic embryo instead of somatic embryogenesis.

Changed accordingly.

  1. L238: A. thaliana (Italic).

Changed accordingly.

  1. L297: Arabidopsis thaliana (Italic).

Changed accordingly.

  1. L465: calli (non-italic).

Changed accordingly.

Reviewer 3 Report

The Authors propose to follow auxin roles and changes during somatic embryogenesis in tamarillo.

Auxin quantification and localization using different methods, as well as gene expression monitoring are proposed.

This work is in my opinion of interest for the Plants reader. The experiments were well conducetd and the ms well prepared.

I have only two very minor suggestions:

Please enlarge Figure 1 or provide better quality pictures

Line 463: "Some genetic changes have also been found in the main auxin response proteins." I think: "Some changes in gene expression level have also been found in some of the main auxin response genes."

Author Response

Dear Editor,

Firstly, we would like to acknowledge the suggestions, corrections and comments of the reviewers that allowed us to improve the manuscript.

            All the comments have been taken into account and several sections of the manuscript have been rewritten, expanded or new figures have been added. The original manuscript lists the changes that we have made in response to the reviewers and in this letter we respond to the comments and corrections of each reviewer. Furthermore, as a new discussion topic was added, and another expanded, the reference list was also modified.

            In the following text our answers to the different reviewers are in blue.

            We hope that the new manuscript has met the standard for publication in Plants.

            Kind regards,

            Jorge Canhoto (on behalf of all authors)

  1. Please enlarge Figure 1 or provide better quality pictures

Figure 1 has been enlarged and slightly changed.

  1. Line 463: "Some genetic changes have also been found in the main auxin response proteins." I think: "Some changes in gene expression level have also been found in some of the main auxin response genes."

Changed accordingly.

Round 2

Reviewer 1 Report

The Authors have corrected and improved the manuscript according to my suggestions. I advise to publish the manuscript after some minor changes pointed in the attached file.

Author Response

Dear Editor,

Once again, the authors would like to thank the reviewers for their valuable comments and suggestions.

The minor reviews presented in the last draft have been corrected. The changes were made on the first corrected abstract and here follows a list of the last changes made. Our answers are in blue.

Jorge Canhoto (on behalf of all authors)

  1. Abstract line 17.

“With” has been removed.

  1. Figure 1.

Legend has been corrected, EC 2 and NEC 2 refer to calli induced from leaf segments as stated in the figure and the caption was incorrect.

  1. Table 1.

The letters referring to the statistical analysis of the number of abnormal somatic embryos at 1 and 10 µM of TIBA have been corrected to “c”.

  1. Figure 3.

(C) refers to statistical analysis in non-embryogenic calli. The legend has been changed accordingly.  

  1. Line 213.

The figure number has been changed. The figure was also incorrectly named in line 216. It has been changed accordingly.   

Reviewer 2 Report

The authors addressed all the issues. I do not have any further comments.

Author Response

No further comments were made by this reviewer